# Prediction model for periodontitis stage based on the salivary microbiome

Jaewoong Lee,[1] Hyun-Joo Kim,[2,3] Eun-Hye Kim,[4,5] Seunghoon Kim,[1] Byeongjun Park,[1] Suji Hong,[5] Jihoon Kang,[5] Ju-Youn Lee,[2,3] Semin Lee[1]

**ABSTRACT** This study aimed to characterize salivary microbiome compositions that can classify periodontal health and various stages of periodontitis. We collected saliva samples from 250 study subjects, including 100 periodontally healthy controls and 150 periodontitis patients in stages I/II/III. We performed 16S ribosomal RNA gene sequencing to characterize their salivary microbiomes. Alpha diversities show significant differences between healthy and periodontitis. Differentially abundant taxa were identified by ANCOM. Random forest machine learning models were used to classify each periodontitis stage based on the centered log-ratio of differentially abundant taxa. We identified 20 differentially abundant taxa among the groups in the salivary microbiomes of all groups. Among these differentially abundant taxa, *Porphyromonas gingivalis* and *Actinomyces* spp. are the most important taxa on the random forest model to classify the periodontitis statuses. Our random forest model classified multiple periodontitis statuses with an area-under-curve of 0.829 ± 0.124, sensitivity 0.884 ± 0.022, and specificity 0.652 ± 0.065. Moreover, because it can be difficult to diagnose in dentistry practice, we performed our classifier model to distinguish healthy or stage I, providing an area-under-curve of 0.736 ± 0.168, sensitivity 0.789 ± 0.102, and specificity 0.622 ± 0.196. Furthermore, our random forest model detected periodontitis patients from healthy individuals with an area-under-curve of 0.924 ± 0.088, sensitivity of 0.862 ± 0.175, and specificity of 0.921 ± 0.061. Finally, we evaluated our classification model with external data sets from Spanish and Portuguese subjects. Some evaluations showed a slight decrease, but it might be due to different salivary microbiome compositions from ethnicity. Significant differences were identified in the differentially abundant taxa among healthy controls and the various stages of periodontitis.

**IMPORTANCE** Periodontitis is a common but complex oral disease that can lead to tooth loss and contribute to systemic health issues. Early and accurate diagnosis is essential for effective intervention, yet traditional diagnostic methods often rely on invasive clinical assessments that may miss early signs. This study demonstrates that salivary microbiome profiles can be used to classify both periodontal health and multiple periodontitis stages using a machine learning approach. By identifying the 20 key microbial taxa, including *Actinomyces* spp., we developed a non-invasive predictive model with high diagnostic accuracy. Importantly, the model was also able to detect early-stage disease and performed well across external data sets, highlighting its potential for broader clinical application. These findings suggest that a salivary microbiome-based diagnostic tool may support more precise, accessible, and early diagnosis of periodontitis in dental disease management.

**KEYWORDS** periodontitis, 16S rRNA gene sequencing, machine learning, salivary microbiome

**Peer Reviewers** Tsute Chen, The Forsyth Institute, Cambridge, Massachusetts, USA; Casey Chen, Ostrow School of Dentistry of USC, Los Angeles, California, USA

Address correspondence to Jihoon Kang, jhkang@helixco.co.kr, Ju-Youn Lee, heroine@pusan.ac.kr, or Semin Lee, seminlee@unist.ac.kr.

Jaewoong Lee, Hyun-Joo Kim, Eun-Hye Kim, and Seunghoon Kim contributed equally to this article as first authors. Author order was determined by consensus among all authors following a review of documented contributions.

The authors declare no conflict of interest.

See the funding table on p. 12.

10.1128/msystems.01103-25 **1**

Periodontitis is a chronic inflammatory disease of the tissue surrounding the tooth, caused by microbial dysbiosis due to plaque accumulation (1). Periodontitis leads to the loss of periodontal attachment and can result in irreversible bone loss and eventually in permanent tooth loss. Following the periodontal disease classification announced in 1999, a new classification of periodontal disease was established in 2018 (2). Under the 2018 classification, diagnosis and staging of periodontitis are determined by clinical measures that primarily reflect accumulated tissue loss, while grading provides context on expected progression and likely treatment response (3). Neither framework directly quantifies real-time disease activity (4). Our study does not measure activity; rather, it evaluates whether salivary microbial profiles can discriminate health and multiple stages of periodontitis, offering candidate biomarkers that may complement staging and grading. Such biomarkers could aid earlier risk stratification and monitoring but require prospective, longitudinal validation before claims about activity or prediction can be made.

Several attempts have been made to replace traditional methods of diagnosing periodontitis, with the use of saliva emerging as a prominent alternative due to the advancement of salivaomics (5). Saliva could be a useful source for periodontitis diagnosis as the collection of saliva is a non-invasive, simple, and patient-friendly source of microbiome. Additionally, most studies have indicated that periodontitis may contribute to the onset or exacerbation of metabolic syndrome (6). Therefore, changes in the levels of these salivary markers can be used as excellent diagnostic, prognostic, and therapeutic biomarkers for periodontitis (7). Periodontitis results from an imbalance between the host defense and the microbial community (8). Although periodontitis is a multifactorial disease influenced by several contributing factors, such as diet, stress, and smoking, the onset of periodontitis inevitably involves both qualitative and quantitative shifts in the microbial community (9, 10). The composition and characteristics of the subgingival microbiome differ according to the periodontal status (9), suggesting that microbiome composition profiling based on clinical diagnostic criteria could serve as a new etiological diagnostic standard. Therefore, many studies have characterized subgingival microbiomes in the context of periodontitis. Recently, some studies have applied high-throughput sequencing techniques to identify differences in the subgingival microbiome composition from different periodontal health statuses (9, 11, 12). This recognition highlighted that changes in the microbiome composition, specifically dysbiosis, are crucial factors in the pathogenesis and development of periodontitis. However, most of these studies either did not consider multiple periodontitis stages, included an insufficient number of study participants, or focused solely on the subgingival microbiome compositions.

Previous studies have developed prediction models based on subgingival microbiome composition for diagnosing periodontitis (13), which have demonstrated high diagnostic accuracy and can be applied to individual saliva samples (14). While these indices provide helpful insights, they are often limited to classifying periodontitis stages. Additionally, many existing machine learning models are trained on only the presence of periodontitis, not on the multiple periodontitis stages.

To contextualize performance, we benchmarked against previous studies (13, 14). The plaque-based subgingival microbial dysbiosis index (SMDI) showed high accuracy for health vs periodontitis (13). A subsequent adaptation to saliva and tongue also achieved strong binary accuracy across oral niches (14). Our study complements these findings by (i) using saliva, a non-invasive matrix suited for screening, and (ii) extending beyond binary dysbiosis to multi-class staging with a compact 20-taxa-based machine learning model. For the binary comparison, our machine learning model attains an area-under-curve (AUC) comparable to reported SMDI values, while additionally providing calibrated class probabilities for stage resolution. Differences in sampling niche (plaque vs saliva), feature construction (index vs supervised panel), and pipelines likely explain residual metric differences.

We recently used the copy number of nine periodontitis-associated pathogens from saliva to predict periodontitis stages using multiplex quantitative PCR and machine learning techniques (15). However, our previous study was limited by focusing on only nine periodontal pathogens and did not address the broader bacterial diversity associated with periodontitis stages. Therefore, this study aims to address these knowledge gaps by developing a machine learning model capable of classifying periodontitis stages based on salivary microbiome compositions, providing more detailed and clinically relevant tools for assessing periodontitis progression. Hence, we performed 16S rRNA gene sequencing to analyze the salivary microbiomes of healthy individuals and patients with periodontitis. Additionally, we aimed to identify biomarkers for the accurate prediction of periodontitis stages using the classification established in 2018 (2).

## RESULTS

### Summary of study participants and sequencing data

A total of 250 participants were enrolled in this study. The patients with periodontitis were categorized as stage I ($n = 50$), stage II ($n = 50$), or stage III ($n = 50$) based on the 2018 classification of periodontitis (2). Table 1 represents the clinical information of the study participants, including age, attachment level, probing depth, plaque index, and gingival index. No significant difference in sex was observed in any stage (Table 1). Periodontal parameters, including attachment level, probing depth, plaque index, and gingival index, were significantly increased with periodontitis stages (Kruskal-Wallis test $P \leq 0.001$).

### Diversity indices reveal differences in diversity among the periodontitis stages

Alpha-diversity indices represent the distribution of taxonomies within each sample. Measures such as ACE, Chao1, Fisher, Margalef, and observed amplicon sequence variants (ASVs) indicated significant statistical differences between the healthy status and periodontitis stages I/II/III (Fig. 1a through e), but no significant differences were observed between the periodontitis stages I/II/III. This underscores the importance of using machine learning techniques to classify the salivary microbiome composition and differentiate between the stages of periodontitis.

Beta-diversity indices demonstrated the taxonomic distances between microbiome communities. The confidence ellipses of the tSNE-transformed Aitchison index indicated distinct distributions among healthy status and periodontitis stages (permutational multivariate analysis of variance [PERMANOVA] $P \leq 0.001$; Fig. 1f). Additionally, the uniqueness of each periodontitis stage was further confirmed by statistically significant differences in distances between stages (MWU test $P \leq 0.05$ and PERMANOVA test $P \leq 0.001$; Fig. 1g through j; Table S1).

**TABLE 1** Clinical characteristics of the study participants[a]

| Index | Healthy | Stage I | Stage II | Stage III | P value |
|---|---|---|---|---|---|
| Age (year) | 33.83 ± 13.04 | 43.30 ± 14.28 | 50.26 ± 11.94 | 51.08 ± 11.13 | 6.18E−17 |
| Gender (male) | 44 (44.0%) | 22 (44.0%) | 25 (50.0%) | 25 (50.0%) | NA |
| Smoking (never) | 83 (83.0%) | 36 (72.0%) | 34 (68.0%) | 29 (58.0%) | NA |
| Smoking (ex) | 12 (12.0%) | 7 (14.0%) | 9 (18.0%) | 10 (20.0%) | NA |
| Smoking (current) | 2 (2.0%) | 7 (14.0%) | 7 (14.0%) | 10 (20.0%) | NA |
| Number of teeth | 28.03 ± 2.23 | 27.36 ± 1.80 | 26.72 ± 2.89 | 25.74 ± 4.34 | 8.07E−05 |
| Attachment level (mm) | 2.45 ± 0.29 | 2.75 ± 0.38 | 3.64 ± 0.83 | 4.54 ± 1.14 | 1.82E−35 |
| Probing depth (mm) | 2.42 ± 0.29 | 2.61 ± 0.40 | 3.27 ± 0.76 | 3.95 ± 0.88 | 6.43E−28 |
| Plaque index | 17.66 ± 16.21 | 35.46 ± 23.75 | 54.40 ± 23.79 | 58.30 ± 25.25 | 3.23E−22 |
| Gingival index | 0.09 ± 0.16 | 0.44 ± 0.46 | 0.85 ± 0.52 | 1.06 ± 0.52 | 2.59E−32 |

[a]Significant differences were assessed using the Kruskal-Wallis test. NA, not applicable.

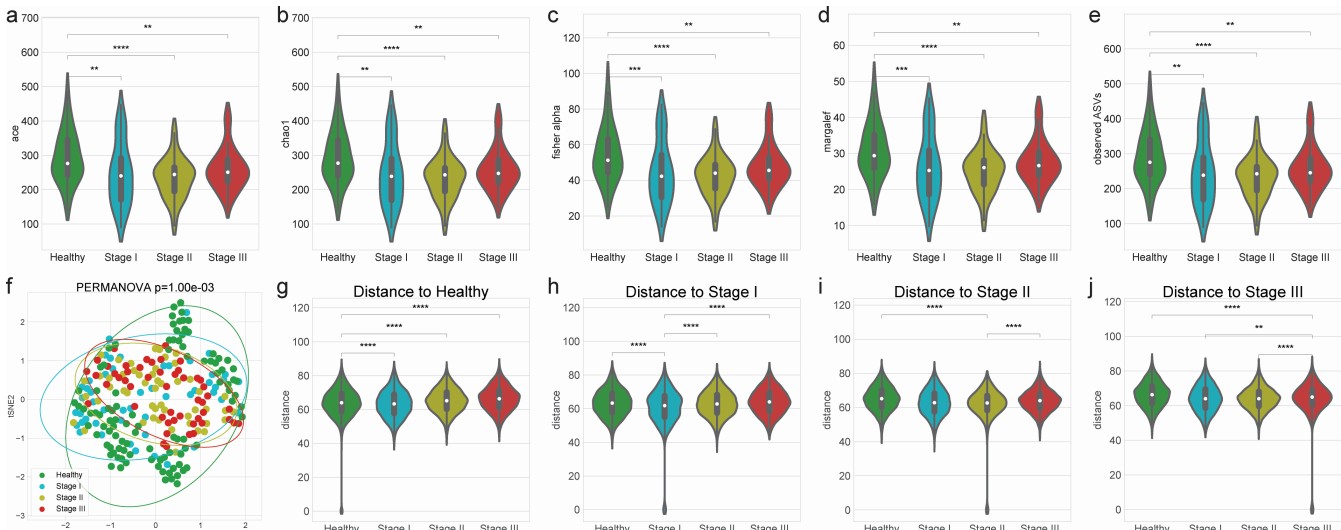

**FIG 1** Diversity indices. Comparisons of salivary microbiomes among healthy controls and patients with periodontitis. Alpha-diversity indices indicate that healthy controls have increased heterogeneity than periodontitis stages as measured by (a) ACE, (b) Chao1, (c) Fisher, (d) Margalef, and (e) observed ASVs. (f) The beta-diversity index (Aitchison index) was visualized using a t-distributed stochastic neighbor embedding (tSNE)-transformed plot. The confidence ellipses are shown to display the distribution of each periodontitis stage. The distance to each stage demonstrated that each periodontitis stage was distinguished from the other periodontitis stages: (g) distance to healthy, (h) distance to stage I, (i) distance to stage II, and (j) distance to stage III. Statistical significance was determined by the Mann-Whitney U-test (MWU): $P < 0.01$ (**) and $P < 0.0001$ (****).

## Differentially abundant taxa among different periodontitis stages

Out of the 425 total identified taxa (Fig. S1), 20 differentially abundant taxa (DAT) among the different periodontitis stages were identified by ANCOM (16). Hierarchical clustering analysis of the sample-level abundances of these 20 DAT grouped the samples into three distinct groups (Fig. 2a). Group 1 comprised 10 taxa that were depleted in the healthy individuals but relatively enriched in stages II/III. This group included *Treponema* spp., *Prevotella* sp. HMT 304, *Prevotella* sp. HMT 526, *Peptostreptococcaceae[XI][G-5] saphenum*, *Treponema* sp. HMT 260, *Mycoplasma faucium*, *Peptostreptococcaceae[XI][G-9] brachy*, *Lachnospiraceae[G-8]* bacterium HMT 500, *Peptostreptococcaceae[XI][G-6] nodatum*, and *Fretibacterium* spp. In addition, the seven taxa of group 2 were significantly enriched in all three stages of periodontitis compared to the healthy status. Group 2 included *Porphyromonas gingivalis*, *Campylobacter showae*, *Filifactor alocis*, *Treponema putidum*, *Tannerella forsythia*, *Prevotella intermedia*, and *Porphyromonas* sp. HMT 285. In contrast, the three taxa of group 3, including *Actinomyces* spp., *Corynebacterium durum*, and *Actinomyces graevenitzii,* were significantly enriched in the healthy status but were depleted in stages II/III. These patterns were further confirmed by the centered log-ratio (CLR) within the 20 DAT (Fig. 2b), indicating that periodontitis is mainly associated with the DAT rather than with other salivary bacteria.

## Classification of periodontitis stages by random forest models

Random forest classifiers were trained to classify periodontitis stages based on the CLR of DAT. First, we performed multilabel classifications for the periodontally healthy control and different periodontitis stages I, II, and III. Using random forest classification, we achieved the highest BA at 0.768 ± 0.043 in classifying different periodontal stages (Table 2). AUC ranged from 68% (for stage I vs other stages) to 90% (for healthy vs other stages) (Fig. 3b; Table 2).

Second, we performed a random forest classification for healthy individuals and patients with stage I periodontitis, since the early diagnosis in general dental practice is challenging (3). Notably, the random forest classification achieved the highest BA at

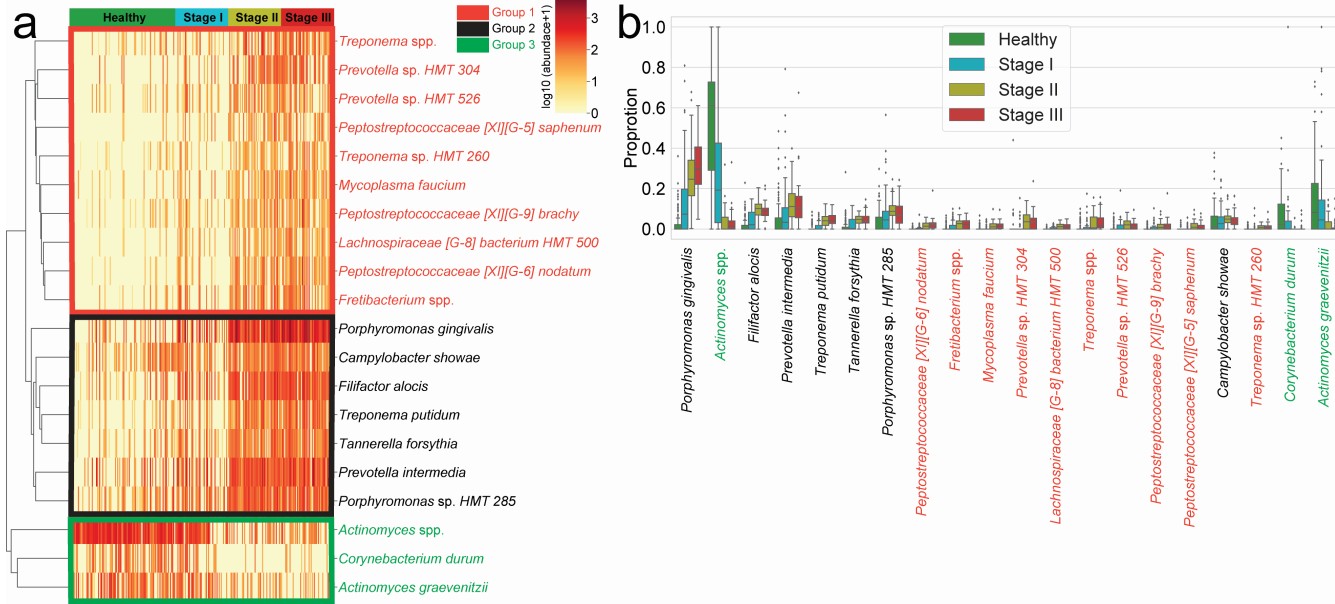

**FIG 2** Differentially abundant taxa. DAT that were identified by ANCOM. (a) Heatmap of clustered DAT with similar distribution among subjects. Group 1, group 2, and group 3 are marked in red, black, and green, respectively. (b) Box plots showing the proportions of DAT. Taxa are ordered by the ANCOM W statistics, that is number of rejected pairwise null hypotheses. Samples are ordered by their attachment level.

0.738 ± 0.152 (Table 2; Fig. 3c). This model demonstrated high sensitivity and specificity for the diagnosis of stage I periodontitis (AUC = 73%) (Fig. 3d).

Next, we performed a multi-label classification by combining stages II and III, based on the observation that patients with stage II periodontitis have microbial compositions more similar to those in stage III than to other statuses (Fig. 1f and j). Random forest classification achieved the highest BA of 0.838 ± 0.049 for the classification of healthy, stage I, and stages II/III groups (Table 2; Fig. 3e). The AUC values ranged from 70% (for stage I vs other stages) to 96% (for stages II/III vs other stages) (Fig. 3f).

Moreover, we performed random forest classification to distinguish between healthy controls and patients with periodontitis (stages I, II, and III). Random forest classification achieved the highest BA at 0.892 ± 0.073 with a high AUC of 92% (Table 2; Fig. 3g).

Finally, to enable direct and cohort-level comparisons using the same DAT panel, we validated our random forest classification using data from Spanish subjects (11) and Portuguese subjects (17) using stratified K-fold cross-validation on the CLR inputs (Fig. 4).

**TABLE 2** Feature combinations and their evaluations[a]

| Classification | Features | ACC | AUC | BA | F1 | PRE | SEN | SPE |
|---|---|---|---|---|---|---|---|---|
| Healthy vs stage I vs stage II vs stage III | *Act.* | 0.740 ± 0.037 | 0.721 ± 0.153 | 0.653 ± 0.049 | 0.827 ± 0.025 | 0.827 ± 0.025 | 0.827 ± 0.025 | 0.480 ± 0.074 |
| | *Act.* + *C. durum* | 0.796 ± 0.037 | 0.785 ± 0.122 | 0.728 ± 0.049 | 0.864 ± 0.024 | 0.864 ± 0.024 | 0.864 ± 0.024 | 0.592 ± 0.073 |
| | Top 16 taxa | 0.826 ± 0.032 | 0.829 ± 0.124 | 0.768 ± 0.043 | 0.884 ± 0.022 | 0.884 ± 0.022 | 0.884 ± 0.022 | 0.652 ± 0.065 |
| Healthy vs stage I | *Act.* | 0.660 ± 0.128 | 0.705 ± 0.166 | 0.607 ± 0.164 | 0.750 ± 0.087 | 0.750 ± 0.081 | 0.762 ± 0.140 | 0.453 ± 0.196 |
| | *Act.* + *A. graevenitzii* | 0.713 ± 0.090 | 0.708 ± 0.168 | 0.690 ± 0.123 | 0.798 ± 0.065 | 0.850 ± 0.092 | 0.758 ± 0.080 | 0.622 ± 0.196 |
| | Top 11 taxa | 0.767 ± 0.120 | 0.736 ± 0.188 | 0.783 ± 0.152 | 0.837 ± 0.090 | 0.900 ± 0.118 | 0.789 ± 0.102 | 0.777 ± 0.243 |
| Healthy vs stage I vs stages II/III | *Act.* | 0.773 ± 0.058 | 0.779 ± 0.169 | 0.745 ± 0.065 | 0.830 ± 0.043 | 0.830 ± 0.043 | 0.830 ± 0.043 | 0.660 ± 0.086 |
| | *Act.* + *C. durum* | 0.808 ± 0.039 | 0.827 ± 0.155 | 0.784 ± 0.044 | 0.856 ± 0.029 | 0.856 ± 0.029 | 0.856 ± 0.029 | 0.712 ± 0.059 |
| | Top 9 taxa | 0.856 ± 0.043 | 0.856 ± 0.154 | 0.838 ± 0.049 | 0.892 ± 0.032 | 0.892 ± 0.032 | 0.892 ± 0.032 | 0.784 ± 0.065 |
| Healthy vs stages I/II/III | *Act.* | 0.784 ± 0.105 | 0.885 ± 0.095 | 0.802 ± 0.103 | 0.737 ± 0.097 | 0.730 ± 0.110 | 0.784 ± 0.178 | 0.819 ± 0.054 |
| | *Act.* + *P. gingivalis* | 0.840 ± 0.112 | 0.911 ± 0.086 | 0.859 ± 0.106 | 0.816 ± 0.109 | 0.830 ± 0.100 | 0.835 ± 0.188 | 0.882 ± 0.057 |
| | Top 16 taxa | 0.864 ± 0.092 | 0.924 ± 0.088 | 0.892 ± 0.073 | 0.846 ± 0.082 | 0.870 ± 0.110 | 0.862 ± 0.175 | 0.921 ± 0.061 |

[a]Classification performance with the most important taxon, the two most important taxa, and taxa with the best-balanced accuracy. *Act.*, *A. graevenitzii*, and *C. durum* are *Actinomyces* spp., *Actinomyces graevenitzii*, and *Corynebacterium durum*, respectively. ACC, accuracy; AUC, area-under-curve; BA, balanced accuracy; PRE, precision; SEN, sensitivity; SPE, specificity.

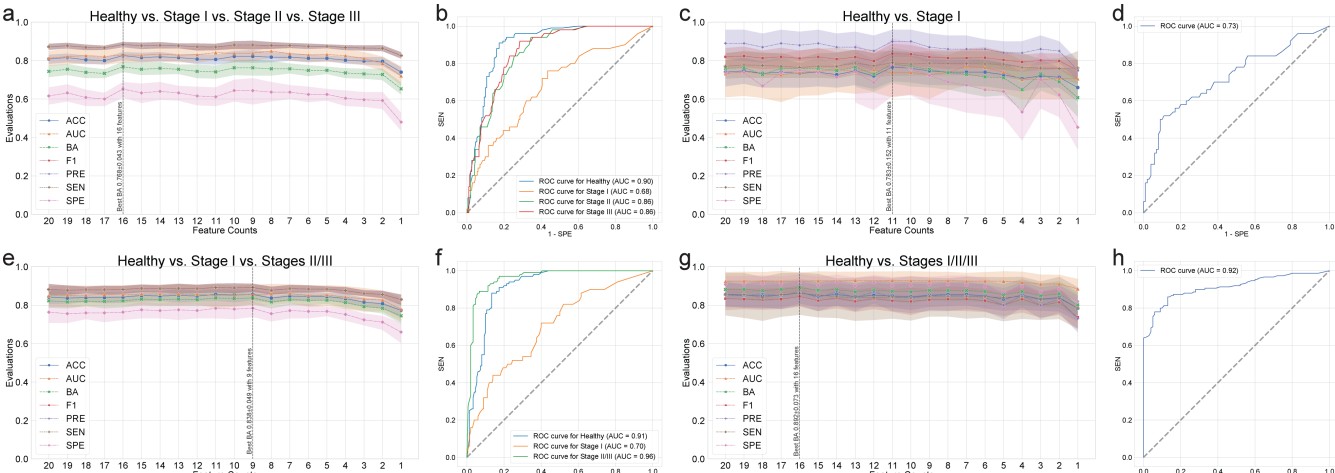

**FIG 3** Random forest classification. The classification metrics in the random forest classifications were as follows: accuracy (ACC), AUC, balanced accuracy (BA), F1 score (F1), precision (PRE), sensitivity (SEN), and specificity (SPE). Every classification metric ranges from [0, 1], with higher values indicating better performance. The feature counts mean that the classification model was trained on the most important *n* features, as in Table S2. (a) Classification performance for healthy vs stage I vs stage II vs stage III. (b) Receiver operating characteristics (ROC) curve for the highest BA of panel a. (c) Classification performance for healthy vs stage I. (d) ROC curve on the highest BA of panel c. (e) Classification performance for healthy vs stage I vs stages II/III. (f) ROC curve for the highest BA of panel e. (g) Classification performance for healthy vs stages I/II/III. (h) ROC curve for the highest BA of panel g.

The multi-label classification model yields comparable ACC, BA, and SPE across cohorts (MWU test $P \geq 0.05$; Fig. 4a), despite uneven stage distributions in the external data sets. Early-stage classifications show the same patterns (Fig. 4b and c), and classification for healthy controls remains high across all cohorts (MWU test $P < 0.05$; Fig. 4d). These results support the portability of the DAT marker set, with any residual variation likely attributable to class imbalance and smaller sample sizes in the external cohorts.

## DISCUSSION

In this study, we conducted a cross-sectional analysis using 16S rRNA gene sequencing to investigate potential alterations in the composition of the salivary microbiome according to periodontitis stages. Periodontitis stage classification was based on the 2018 periodontitis classification (2). Significant differences in the composition of salivary microbiomes between different periodontitis statuses were observed. Additionally, our random forest classifier predicted different periodontitis statuses with a high AUC of $0.870 \pm 0.079$ using the CLR of DAT between the study participants (Table 2).

A previous study selected the major causative agents of periodontitis, the red complex, which includes *T. forsythia*, *P. gingivalis*, and *Treponema denticola* (18). However, other studies have indicated that periodontal pathogens participate in microbiome networks with other oral bacteria to construct dental plaque before the progression of periodontitis (19–21). Recent studies have demonstrated a relationship between the oral microbiome composition and periodontitis severity using subgingival samples (11, 12, 22). Building on prior efforts, we have analyzed the microbial composition in the saliva from periodontally healthy controls and patients with different stages of periodontitis.

Our results identified 425 taxa in the salivary microbiome (Fig. S1). To calculate the heterogeneity within each salivary microbiome community, we estimated the alpha-diversity indices (ACE, Chao1, Fisher, Margalef, and observed ASVs). Our results have demonstrated that periodontally healthy controls exhibited a higher species richness than did patients with periodontitis, as indicated by the alpha-diversity indices (Fig. 1a through e). These findings are consistent with those of reports showing less diverse communities in patients with aggressive periodontitis than in periodontally healthy populations (23). Considering the increase in *P. gingivalis* abundance with multiple periodontitis stages, the salivary microbiomes maintained microbial networks

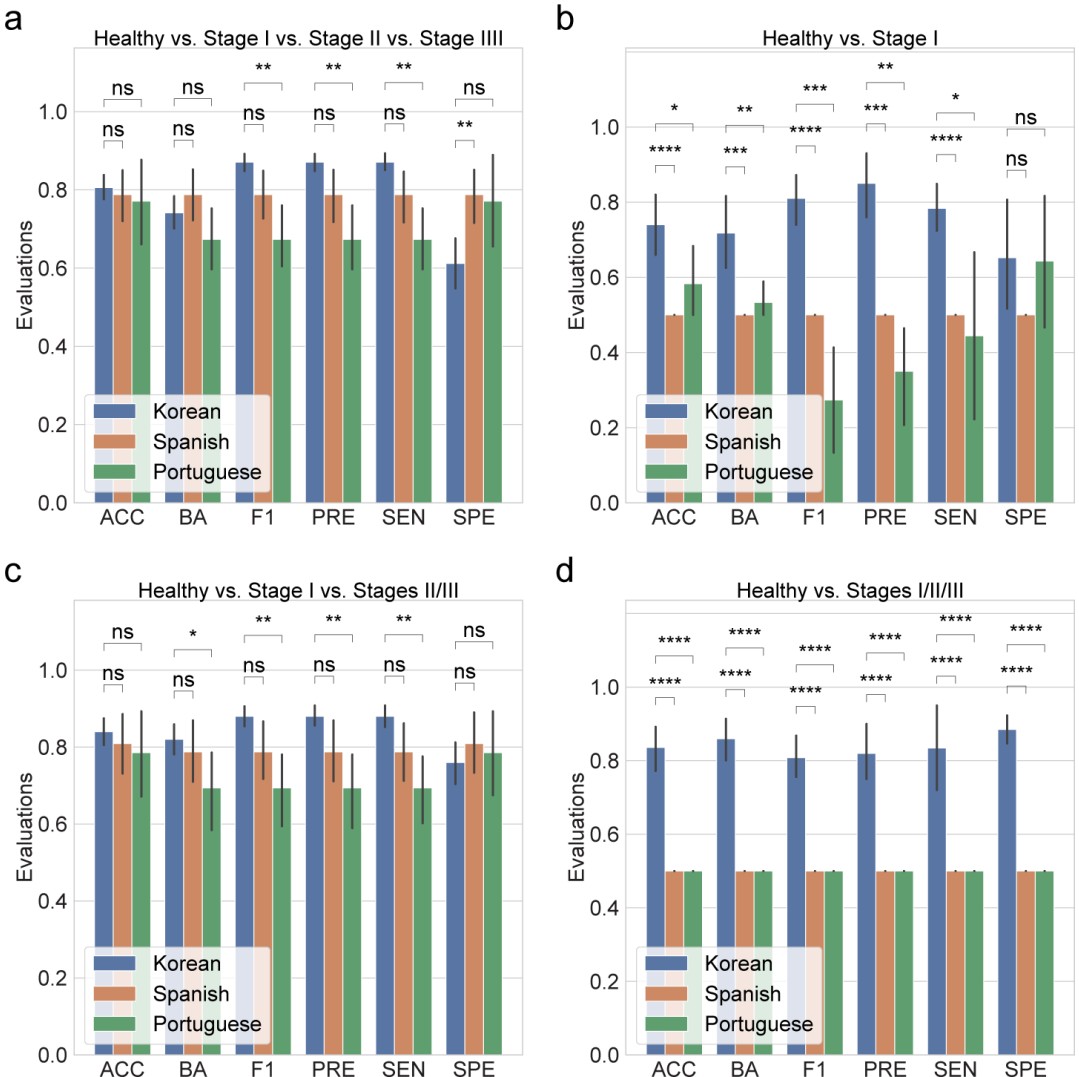

**FIG 4** Random forest classification with validation datasets. The classification metrics in the random forest classification included ACC, BA, F1, PRE, SEN, and SPE. Every classification metric ranges from [0, 1], with higher values indicating better performance. (a) Classification performance for healthy vs stage I vs stage II vs stage III. (b) Classification performance for healthy vs stage I. (c) Classification performance for healthy vs stage I vs stages II/III. (d) Classification performance for healthy vs stages I/II/III. Statistical significance was determined by the MWU test: $P \geq 0.05$ (ns), $P < 0.05$ (*), $P < 0.01$ (**), $P < 0.001$ (***), and $P < 0.0001$ (****).

dominated by *Streptococcus* spp. in periodontally healthy controls. However, several periodontal pathogens, including *P. gingivalis*, can induce dysbiosis in the salivary microbiomes, leading to the pathogenesis of periodontitis. However, a previous study reported that patients with periodontitis had a higher alpha-diversity index (observed ASVs) than healthy controls in the subgingival microbiome (11). This discrepancy could be attributed to the different sampling sites: saliva and subgingival plaque. Recent research has reported significant differences in alpha-diversity indices among subgingival plaque, saliva, and tongue biofilms from patients with periodontitis and periodontally healthy controls, with the highest alpha-diversity in saliva samples (24). Furthermore, stage I periodontitis did not demonstrate significant differences in alpha-diversity indices compared to stages II/III periodontitis (Fig. 1a through e). This suggests that stage I periodontitis samples exhibit various heterogeneity reflecting an intermediate state between a healthy status and stages II/III periodontitis. Similarly, most periodontal pathogens are frequently identified at low abundance in gingivitis (9). Additionally, *P. gingivalis* and *T. forsythia* are more frequently detected in patients with early

periodontitis than in healthy controls (25), and in early periodontitis patients with active disease than in inactive subjects (26). To examine the dissimilarity between the study groups, we calculated beta-diversity indices. The distances for each periodontitis stage, including healthy status and stages I/II/III (Fig. 1g through j; Table S1), displayed significant differences among the periodontitis stages. This is consistent with a previous study (11), suggesting that the salivary microbiome composition might have stage-specific characteristics. Periodontitis is a progressive disease, and once the attachment level is lost, it is almost impossible to completely regenerate it. Considering this, being able to simply screen for the early stages of periodontitis through saliva is exceptionally valuable for efficient control of the disease.

Among the total of 425 identified taxa in the salivary microbiome, the 20 DAT that demonstrated significant changes in abundance among periodontally healthy controls and different periodontitis stages were selected using ANCOM (16). Hierarchical clustering split the DAT into three groups (Fig. 2a). Notably, among the red complex pathogens (27), *P. gingivalis* and *T. forsythia* were categorized in group 2 and were more abundant in stages II/III than in healthy individuals. Furthermore, among the orange complex pathogens (28), *C. showae* was also categorized in group 2. Additionally, some of the DAT belonging to group 2, including *F. alocis* (29), *T. putidum* (30), *T. forsythia* (31, 32), and *P. intermedia* (33), have confirmed their critical roles in periodontitis. This suggests that DAT in group 2 plays a crucial role in both the periodontitis pathogenesis and progression of periodontitis. Lafaurie et al. reported significant differences in the proportion of some group 1 DAT, including *Peptostreptococcaceae[XI][G-5] saphenum*, *Peptostreptococcaceae[XI][G-6] nodatum*, and *Peptostreptococcaceae[XI][G-9] brachy*, between healthy individuals and patients with periodontitis (10). These results align with our findings and suggest that group 1 DAT also plays a crucial role in the pathogenesis and development of periodontitis. In contrast, group 3 DAT, including *C. durum* and *A. graevenitzii*, was enriched in healthy controls compared to patients with periodontitis, which is in line with previous studies (34, 35).

We also developed random forest-based classification models for four different classification settings (healthy vs stage I vs stage II vs stage III, healthy vs stage I, healthy vs stage I vs stages II/III, and healthy vs stages I/II/III) to predict different periodontitis stages (Fig. 3; Table 2). *Actinomyces* spp. was the most important feature in all four different classification settings (Table S2). This is consistent with a recent study describing *Actinomyces* spp. as the most dominant taxa in the healthy and gingivitis groups (12). Previously, our group developed periodontitis classification machine learning models to predict the severity of chronic periodontitis, based on the copy numbers of nine salivary bacteria. We classified healthy controls and chronic periodontitis with an AUC of 94%, BA of 84%, SEN of 95%, and SPE of 72%, using six bacterial combinations and a random forest model (15). Na et al. also developed a machine-learning model for classifying periodontitis using 266 species from the buccal microbiome. Their model achieved an AUC of 92%, BA of 84%, SEN of 94%, and SPE of 74% (36). Our machine learning models outperformed previously published models, achieving an AUC of 0.864 ± 0.092, BA of 0.924 ± 0.088, SEN of 0.862 ± 0.175, and SPE of 0.921 ± 0.061, using only 16 DAT to distinguish patients with periodontitis from healthy controls (Fig. 3d; Table 2; Table S2). This indicated that our model improved BA by at least 5% and SPE by at least 17% by identifying group 3 bacteria that were significantly enriched in healthy controls.

To validate the consistency of our random forest classification model, we have tested our prediction model based on publicly available 16S rRNA gene sequencing data from Spanish subjects (11) and Portuguese subjects (17) (Fig. 4), despite the fact that the sequencing data from the Spanish subjects were built on subgingival plaque, not saliva. A limitation of this study is that the classification models were developed and validated primarily on Korean study participants, which may limit their applicability to other ethnic groups with differing salivary microbiome compositions (37, 38). Population-specific variations can impact classification model performances, emphasizing the need for further validation and adaptation across diverse ethnic backgrounds.

Our external validation included a Spanish cohort profiled from subgingival plaque rather than saliva. We selected this data set for its stage-specific diagnoses, but the matrix mismatch introduces domain shifts (e.g., niche-specific communities, biomass, biofilm architecture) and protocol heterogeneity (e.g., lysozyme and bead-beating), which plausibly explains the attenuation in specificity and related metrics. We intentionally did not retrain on plaque to isolate the transportability of a saliva-trained model; these results should be interpreted as a stringent cross-niche stress test rather than a like-for-like replication. Although all cohorts were reprocessed with a harmonized pipeline and evaluated on the same 20-DAT panel, residual matrix and batch effects likely remain. Future work will prioritize saliva-based external cohorts with matched collection and extraction methods and assess domain-adaptation or transfer-learning strategies, multi-matrix models, and incorporation of clinical covariates.

This study has several limitations acknowledged regarding the clinical parameters and potential confounders affecting the analysis of salivary microbiome compositions related to periodontitis status. While we collected data on plaque index, gingival index, attachment level, and probing depth, we did not provide data on the percentage of teeth with a probing depth above a certain threshold, dental furcation involvement, or the percentage of bleeding on probing. This may have limited the comprehensive and detailed provision of information regarding periodontal health. Additionally, the relatively wide age range may hinder the interpretation of the association between age and periodontitis status, emphasizing the need for future research to consider more extensive clinical parameters related to periodontitis. Furthermore, potential confounders beyond systemic diseases (e.g., body mass index), smoking status (e.g., e-cigarette usage), and other oral-condition variables (e.g., active caries, salivary flow rate) may have influenced oral health and microbiome composition. Incorporating these factors in future studies would provide a more comprehensive understanding of the interplay between lifestyle variables and their impact on oral microbiome composition and periodontal health. Overall, addressing these limitations will enhance our knowledge and improve future research in this field, providing valuable insights into the association between the salivary microbiome and systemic conditions.

Another limitation of this study was that our DNA extraction method did not include lysozyme, a reagent that primarily acts on gram-positive bacteria, because bacteria associated with periodontal diseases are known to be primarily gram-negative anaerobes (39). However, we were able to identify some gram-positive bacteria in our data, including *Peptostreptococcaceae[XI][G-6] nodatum* and *F. alocis*, which emerged as important features in our prediction models. If DNA extraction had been performed using enzymes, such as lysozyme and/or lysostaphin, we might have revealed a broader spectrum of bacteria, encompassing both gram-positive and gram-negative species associated with periodontitis.

## MATERIALS AND METHODS

### Study participants

A total of 250 subjects who were enrolled in this study (100 healthy controls and 150 subjects with periodontitis) visited the Department of Periodontics at Pusan National University Dental Hospital between August 2018 and March 2019. The 150 patients with periodontitis were equally divided into three periodontitis stages: stage I, stage II, and stage III. Exclusion criteria are as follows: (i) those who received periodontal treatment (scaling and root planing) within the past 6 months; (ii) those with systemic diseases, such as uncontrolled diabetes, that can affect the progression of periodontitis; (iii) those who took antibiotics or anti-inflammatory drugs in the preceding 3 months; (iv) women who were pregnant or breastfeeding; (v) those with acute infection (e.g., herpetic gingivostomatitis) or chronic mucosal lesions (e.g., pemphigus or pemphigoid); (vi) those who refused to sign the informed consent form.

## Clinical procedure

All the clinical examinations were performed by an experienced periodontist (H.-J. K.). Probing depth and gingival recession were measured at six sites per tooth (mesiobuccal, midbuccal, distobuccal, mesiolingual, midlingual, and distolingual) for all teeth. For measurements at each tooth site, a periodontal probe (Hu-Friedy, USA) was positioned parallel to the long axis of the tooth at each site. Clinical attachment level was measured from the cementoenamel junction of the tooth, and periodontal pocket depth was measured from the marginal gingival level of the tooth to the deepest point of probing. Plaque index was measured by probing four surfaces per tooth (mesial, distal, buccal, and palatal or lingual) for all teeth. We scored from 0 to 3: 0, no plaque present; 1, a film of plaque adhering to the free gingival margin and adjacent area of the tooth. The plaque may only be recognized by using a probe on the tooth surface; 2, moderate accumulation of soft deposits within the gingival pocket, or the tooth and gingival margin, which can be seen with the naked eye; and 3, abundance of soft matter within the gingival pocket and/or on the tooth and gingival margin. The plaque index of each individual was determined as the arithmetic mean of the plaque indices obtained for each tooth. The gingival index was measured by assessing gingival bleeding by probing all four surfaces of each tooth. The gingiva was scored based on a scale from 0 to 3: 0, normal gingiva, with no inflammation or discoloration; 1, mild inflammation, with slight color changes and minimal edema, but no bleeding on probing; 2, moderate inflammation, with redness, edema, glazing, and bleeding on probing; and 3, severe inflammation, with marked redness, significant edema, ulceration, and spontaneous bleeding. The gingival index of each individual was determined as the arithmetic mean of the gingival indices obtained for each tooth. Although bleeding on probing and furcation involvement were comprehensively considered in the diagnosis and staging process, the relevant data were not shown.

Periodontitis was diagnosed according to the criteria outlined by the 2017 World Workshop on the Classification of Periodontal and Peri-Implant Diseases and Conditions (2, 40). Periodontal health status was defined as <10% bleeding sites with probing depth ≤3 mm. An experienced periodontist (H.-J.K.) classified the stages of periodontitis by considering both severity and complexity, based on clinical examinations using a periodontal probe and radiographic images. Periodontitis is categorized into stage I, stage II, or stage III based on the following criteria.

Stage I was as follows: (i) interdental clinical attachment level at the site of greatest loss: 1–2 mm; (ii) radiographic bone loss: coronal third (15%); (iii) tooth loss due to periodontitis: none.

Stage II was as follows: (i) interdental clinical attachment level at the site of greatest loss: 3–4 mm; (ii) radiographic bone loss: coronal third (15%–33%); (iii) tooth loss due to periodontitis: none.

Stage III was as follows: (i) interdental clinical attachment level at the site of greatest loss: ≥5 mm; (ii) radiographic bone loss: extending to the mid-third of the root and beyond; (iii) tooth loss due to periodontitis: ≤4 teeth.

Complexity was assessed considering maximum probing depth (≤4 mm for stage I, ≤5 mm for stage II, and ≥6 mm for stage III), bone loss pattern (horizontal and vertical bone loss), and dental furcation involvement, such as class II or III for stage III. We included only generalized forms (≥30% of teeth involved) of periodontitis.

For saliva sampling, all subjects were instructed to refrain from consuming food and drink, brushing, or using mouthwash for at least 1 h before the procedure. Sampling was scheduled for saliva sampling between 09:00 a.m. and 11:00 a.m. Mouth rinse was collected by rinsing the mouth for 30 s with 12 mL of a solution (E-zen Gargle; JN Pharm, South Korea). The samples were labeled with the subject's ID and stored at 4°C.

Genomic DNA was extracted from mouthwash samples using an Exgene Clinic SV DNA extraction kit (GeneAll, Seoul, South Korea), and DNA quality and quantity were assessed using a NanoDrop spectrophotometer (Thermo Fisher Scientific, Wilmington, DE, USA). The V3–V4 hypervariable regions of the 16S rRNA genes were amplified using

the forward primer (5′-TCGTCGGCAGCGTCAGATGTGTATAAGAGACAGCCTACGGGNGGCW GCAG-3′) and reverse primer (5′-GTCTCGTGGGCTCGGAGATGTGTATAAGAGACAGGACTAC HVGGGTATCTAATCC-3′). Libraries were prepared according to the standard guidelines of the Illumina 16S Metagenomic Sequencing Library Preparation protocol. The PCR conditions for the first step were as follows: heat activation at 95℃ for 3 min, followed by 25 cycles of 95℃ for 30 s, 55℃ for 30 s, and 72℃ for 30 s. For the final library construction, 10 µL of the purified first PCR products were amplified with Nextera XT Indexed Primer. The second PCR used the same conditions as the first, but with 10 cycles. 16S rRNA sequencing was performed via $2 \times 300$ bp paired-end sequencing at Macrogen Inc. (Macrogen, Seoul, Korea) using the Illumina MiSeq platform (Illumina, San Diego, CA, USA).

## Bioinformatics analysis

The 16S rRNA sequences from the study participants were imported into QIIME2 (version 2020.8) with default parameters for further processing (41). Each sequence was demultiplexed and filtered for quality using DADA2 (42). Thereafter, we assigned high-quality ASVs to exact sequence matches, and taxonomy was estimated using the Human Oral Microbiome Database (version 15.21) (43) and scikit-learn (44). After the taxonomy assignment, ASVs with identical taxonomic classification were collapsed by summing abundances.

To measure the divergence of phylogenetic information in each sample, we calculated alpha-diversity indices for taxonomic richness within a single community and beta-diversity indices for taxonomic differences among different communities. Five alpha-diversity indices, including the ACE, Chao1, Fisher, Margalef, and observed ASVs indices, were calculated using the scikit-bio Python package (version 0.5.5). Additionally, the alpha-diversity indices were compared pairwise using the MWU test.

Furthermore, a beta-diversity (Aitchison index) was calculated using QIIME2. To visualize multidimensional data from the beta-diversity index calculation, we used the t-SNE algorithm, a dimensionality reduction algorithm that transforms high-dimensional data into low-dimensional data. Moreover, to display brief distributions of the data points in tSNE-transformed data, we drew confidence ellipses with 2−σ standard deviations that enclose approximately 90% of the data points. To measure statistical differences among periodontitis stages, distances to each periodontitis stage were selected and compared using the PERMANOVA test and the MWU test.

DAT between subjects with different periodontal health statuses were identified at the genera and species level by ANCOM. Unlike traditional methods that analyze raw counts, ANCOM accounts for the salivary microbiome composition data by comparing log-ratios between taxa. This approach reduces the risk of false positives due to the sum constraint inherent in compositional data. To characterize sub-groups of DAT that have comparable abundance patterns on periodontitis stages, hierarchical clustering was performed on the log-transformed absolute abundances of DAT. Moreover, to minimize other salivary bacteria that have non-significant differences among different periodontitis stages, we analyzed the CLR among the DAT.

The random forest classifier is one of the most powerful and stable classification algorithms for multi-label classification problems (45). To obtain consistent and reliable classification results (46), we performed stratified k-fold cross-validation (k = 10) by periodontitis stage. Furthermore, to determine which features maximize classification evaluations and minimize sequencing efforts, we evaluated the classification results using some features with confusion matrices and their derivations. We applied the backward elimination method to iteratively remove the least important taxa from the input features of the random forest classification model, using the 20 DAT identified through ANCOM analysis. The first step was to calculate the feature importance of each taxon using random forest classification. Then, we sequentially removed bacteria with the lowest importance. The multi-label confusion matrix was transformed into multiple

binary-label confusion matrices using the one-versus-rest method. The confusion matrix was evaluated for ACC, AUC, BA, F1, PRE, SEN, and SPE.

## External data validation

To validate the consistency of our random forest classification, we utilized external data sets from Spanish subjects (accession ID PRJNA863881) (11) and Portuguese subjects (accession ID PRJNA623352) (17). The external data were processed using the same pipeline and parameters as those applied to our study population to ensure reproducibility and reliability. Specifically, the raw 16S rRNA gene sequencing data from these cohorts were quality-filtered, denoised, and clustered into ASVs following the same bioinformatics workflow used for our data set. We established a minimum sequencing depth threshold of 3,614 as the smallest sample sequencing depth to ensure comparable data quality across data sets. This consistent approach allowed us to directly compare the microbial profiles between our study participants and the external validation subjects.

## ACKNOWLEDGMENTS

We would like to thank Professor Han-Na Kim from Sungkyunkwan University for her thoughtful comments on our analysis.

This study was supported by the Start-up Leap Package Support Project (10302029) funded by the Small and Medium Business Administration (Republic of Korea) and partly by the National Research Foundation of Korea funded by the Ministry of Education (RS-2018-NR031072), the Ministry of Science and ICT (RS-2021-NR058886, RS-2023-00225255, and RS-2023-00261820), and the Ministry of Health and Welfare (RS-2024-00466776).

## AUTHOR AFFILIATIONS

[1]Department of Biomedical Engineering, Ulsan National Institute of Science and Technology (UNIST), Ulsan, Republic of Korea
[2]Department of Periodontology, Dental and Life Science Institute, School of Dentistry, Pusan National University, Yangsan, Republic of Korea
[3]Department of Periodontology and Dental Research Institute, Pusan National University Dental Hospital, Yangsan, Republic of Korea
[4]School of Pharmacy, Jeonbuk National University, Jeonju, Republic of Korea
[5]Helixco Inc., Ulsan, Republic of Korea

## AUTHOR ORCIDs

Jaewoong Lee http://orcid.org/0000-0002-4102-280X
Hyun-Joo Kim http://orcid.org/0000-0001-7553-6289
Eun-Hye Kim http://orcid.org/0000-0002-9225-5865
Seunghoon Kim http://orcid.org/0000-0002-2475-9551
Ju-Youn Lee http://orcid.org/0000-0002-0772-033X
Semin Lee http://orcid.org/0000-0002-9015-6046

## FUNDING

| Funder | Grant(s) | Author(s) |
| --- | --- | --- |
| Small and Medium Business Administration | 10302029 | Suji Hong |
| | | Jihoon Kang |
| National Research Foundation of Korea | RS-2018-NR031072, RS-2021-NR058886 | Semin Lee |

| Funder | Grant(s) | Author(s) |
|---|---|---|
| National Research Foundation of Korea | RS-2023-00225255, RS-2023-00261820, RS-2024-00466776 | Semin Lee |

## AUTHOR CONTRIBUTIONS

Jaewoong Lee, Formal analysis, Investigation, Methodology, Writing – original draft | Hyun-Joo Kim, Conceptualization, Data curation, Writing – review and editing | Eun-Hye Kim, Conceptualization, Data curation, Writing – review and editing | Seunghoon Kim, Formal analysis, Writing – original draft | Byeongjun Park, Formal analysis, Investigation, Visualization | Suji Hong, Funding acquisition, Project administration | Jihoon Kang, Funding acquisition, Project administration, Supervision | Ju-Youn Lee, Conceptualization, Data curation, Supervision, Writing – review and editing | Semin Lee, Conceptualization, Project administration, Supervision

## DATA AVAILABILITY

The data sets generated during and analyzed during the current study are available in the NCBI BioProject under accession PRJNA976179. The Docker image that supports the findings of this study is openly available on Docker Hub at https://hub.docker.com/r/fumire/periodontitis_16s. All codes that support the findings of this study are openly available on GitHub at https://github.com/CompbioLabUnist/Periodontitis_16S.

## ETHICS APPROVAL

The study protocol was approved by the Institutional Review Board of the Pusan National University Dental Hospital (IRB No. PNUDH-2016-019). All participants received complete information regarding the objectives and procedures of this study and provided written informed consent.

## ADDITIONAL FILES

The following material is available online.

### Supplemental Material

**Figure S1 (mSystems01103-25-s0001.pdf).** Absolute abundance of salivary bacterial taxa in the different periodontal statuses at the species level.
**Table S1 (mSystems01103-25-s0002.pdf).** Beta-diversity pairwise comparisons on the periodontitis stages.
**Table S2 (mSystems01103-25-s0003.pdf).** Feature importance of taxa in the classification of different periodontitis stages.
**Table S3, part 1 (mSystems01103-25-s0004.xlsx).** Read-count table for all data sets, Korean cohort #1.
**Table S3, part 2 (mSystems01103-25-s0005.xlsx).** Read-count table for all data sets, Korean cohort #2.
**Table S3, part 3 (mSystems01103-25-s0006.xlsx).** Read-count table for all data sets, Portuguese cohort.
**Table S3, part 4 (mSystems01103-25-s0007.xlsx).** Read-count table for all data sets, Spanish cohort.
**Caption (mSystems01103-25-s0008.txt).** Caption for Table S3.

### Open Peer Review

**PEER REVIEW HISTORY (review-history.pdf).** An accounting of the reviewer comments and feedback.

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
