## [Reviewer comments · mSystems]

Prediction model for periodontitis stage based on the salivary microbiome

Jaewoong Lee, Hyun-Joo Kim, Eunhye Kim, Seunghoon Kim, Byeonjun Park, Suji Hong, Jihoon Kang, Ju-Youn Lee, and Semin Lee

Corresponding Author(s): Semin Lee, Ulsan National Institute of Science and Technology

Review Timeline:

Submission Date:	July 23, 2025
Editorial Decision:	October 1, 2025
Revision Received:	December 5, 2025
Editorial Decision:	December 19, 2025
Revision Received:	December 24, 2025
Accepted:	January 8, 2026

Editor: Katherine Maki

Reviewer(s): Disclosure of reviewer identity is with reference to reviewer comments included in decision letter(s). The following individuals involved in review of your submission have agreed to reveal their identity: Tsute Chen (Reviewer #1); Casey Chen (Reviewer #2)

Transaction Report:

DOI: <https://doi.org/10.1128/msystems.01103-25>

Re: mSystems01103-25 (Prediction model for periodontitis stage based on the salivary microbiome)

Dear Prof. Semin Lee:

Revision Guidelines

Sincerely,
Katherine Maki
Editor
mSystems

Reviewer #1 (Comments for the Author):

Please see attached document

Reviewer #2 (Comments for the Author):

Prediction model for periodontitis 1 stage based on the salivary microbiome

The authors examined the saliva microbiome of 100 periodontally healthy controls and 150 patients with periodontitis in stages I-III using 16S rDNA profiling. The study design and approach were conventional and appropriate. Overall, the study was executed competently, and the manuscript is well-written. While the study is not innovative, it provides solid data and represents an incremental advance in oral microbiome and periodontology.

Comments:

1. The following statement in the Introduction is somewhat confusing. The current diagnosis and classification system, as the authors pointed out, is based on "the damage already caused by the disease." The grading of periodontitis (not included in the study) does offer some information on future disease progression and treatment response. However, neither the diagnosis and classification system nor the current study provides information about "the current disease activity." The authors should revise the paragraph.

"Despite this update, diagnosing periodontitis is still primarily based on clinical indicators of disease progression (2). However, these tools tend to reflect the damage already caused by the disease rather than its current state. Some individuals are more susceptible to developing periodontitis and are more prone to progression to severe generalized periodontitis. Therefore, the framework for diagnosing periodontitis introduced in 2018 also anticipates the future establishment of biomarkers to improve diagnosis and management (3). To enable timely treatment through early intervention, it is necessary to introduce a new etiological indicator based on the current disease activity rather than merely relying on periodontitis progression."

2. In the discussion, it would be interesting to compare the diagnostic performance of the current study to that using SMDI cited in references 13 and 14.

3. The authors should provide information on other subjects' oral conditions, such as caries, restorations, saliva flow, etc., which affected the saliva microbiome.

4. The authors should include the distribution and extent of the disease (localized, generalized, molar/incisor) and grading in the diagnosis.

5. Figure 1g-j should be revised because of the redundancy and the mislabeling of the figure heading. For example, Fig. 1g "Distance to Healthy" compares the distance of all against all and not just each diagnosis against healthy. The authors may delete Fig. 1h-j if Fig. 1g has all the necessary information.

Summary:

The novelty of this manuscript is that Random Forest training on saliva data has not been done before. Thus a report of such topic warrants a place in publications to advance the research in this field. However the report can be improved by additional comparisons and model training, which shouldn't take too long. My questions and suggestions are below:

Major questions/suggestions:

1. Page 19 line 408: "The taxonomies of the tied hits were combined." – I suppose this means tied hits to different species? How were they combined? Was a single species chosen for these multi-specie tied hits? If so the abundance distribution may be biased. Perhaps they should be discarded?
2. Please describe more about the two external datasets, the Spanish and Portuguese ones, i.e., BioProject IDs, how many samples in each stage, which 16S rRNA regions were sequences (V3V4?), sequencing platform etc, as these may affect the differences in the microbiome profiles and consequently validation accuracy.
3. Continuing the questions above, my understanding is that the Spanish dataset was from subgingival plaques, not saliva. This makes this dataset not ideal for validation.
4. Further continuing from above, why not perform trainings for all 3 datasets (Korean, Spanish and Portuguese) both separately and combined. For separate training, the individual dataset can be used to validate the other two datasets. If the training model performs best in the combined data sets, the results can be stated as such and the combined trained model can always be validated with future samples. The model trained with the combined datasets may be more universally applicable and can continue to be trained including more datasets in the future.
5. Were the 20 DAT species also identified as DAT in the Spanish /Portuguese cohort? If not this may help explain the lower ACC. I suggest to identify DAT in these two external datasets, separately and combined with the Korean dataset as well, as suggested above.
6. What was the final RF model trained on? The 20 DAT only? What if the Spanish/Portuguese datasets missing any of these 20 DAT? In my opinion all species identified in the training set should be used for training, not just the 20 DAT. This will minimize the number of missing data (due to not being identified as DAT) in the test samples.

7. Continuing the question about, if the two external datasets do not have the same DAT identified, did you provide zeros for these specific DATs (I hope not).

8. What kind of data were used for RF training? Raw read count? Percent abundance? Or central-log ratios (CLR as used by ANCOM). I would like to see the comparison of at least percent abundance and CLR, as microbiome data are compositional and so CLR will overcome that limit.

Minor questions/suggestions:

1. Which version of ANCOM was used for DAT analysis? There are the original ANCOM, then ANCOM-BC and most recently ANCOMBC2. I hope at least ANCOM-BC was used due to the bias correction feature.

2. Please provide both the ASVs and taxonomy assignment read count tables as supplement files for all 3 datasets.

3. Should perform alpha-, beta-, and DAT analysis Spanish/Portuguese datasets as well.

4. What abundance data were used for alpha and beta diversity calculation? Percent abundance or CLR?

5. The heatmap in Figure 2 should be presented with two-way clustering to show whether the sample profiles cluster together along the 4 stages.

6. Figure 2b, the legend (page 26 line 639) describes that "Taxa were sorted by their importance according to ANCOM" I do not know ANCOM generates importance score. Did you mean p value? Or importance scores from RF?

7. Overall, the microbiome data should be analyzed as CLR not percent abundance, whether in differential abundance or diversity analysis, you may get better results.

8. The word "proportion" used in many times in the manuscript, does not have clear definition. "relative abundance" can be used and defined in the first occurrence as "percent relative abundance".

December 3, 2025

Dear Editors and Reviewers,

Thank you for allowing us to submit a revised draft of our manuscript, titled “**Prediction model for**
**periodontitis stage based on the salivary microbiomes**”, for publication as a *Research Article* in the
*mSystems*. We appreciate the time and effort that the editors and the reviewers dedicated to providing
feedback on our manuscript and are grateful for the insightful comments and valuable improvements
to our paper. Those changes are highlighted in red in the revised manuscript. Please see below our
point-by-point responses to the reviewer’s comments and concerns.

Sincerely,

**Semin Lee, Ph.D.**

Associate Professor

Department of Biomedical Engineering

Ulsan National Institute of Science and Technology (UNIST)

50, UNIST-gil, Eonyang-eup, Ulju-gun, Ulsan, Republic of Korea

seminlee@unist.ac.kr

**Reviewer #1**

**Summary**

The novelty of this manuscript is that Random Forest training on saliva data has not been done before.
Thus, a report on such a topic warrants a place in publications to advance the research in this field.
However, the report can be improved by additional comparisons and model training, which shouldn't
take too long. My questions and suggestions are below.

: Thank you for the positive assessment. We agree that broader comparisons will strengthen our
research. These updates improve transparency and rigor while preserving the scope you support.

**Major comment #1**

Page 19 line 408: "The taxonomies of the tied hits were combined."—I suppose this means tied hits to
different species? How were they combined? Was a single species chosen for these multi-species tied
hits? If so, the abundance distribution may be biased. Perhaps they should be discarded?

: We appreciate you for raising this question. We did not combine ASVs with tied hits to different
species. Rather, we collapsed ASVs that shared the same taxonomic classification, i.e., summed
abundances for different ASVs assigned to the same species, to build the taxon-level feature table.
When assignments were ambiguous across species, we retained the feature at the deepest
unambiguous rank, e.g., genus, and did not force a single-species choice. We revised the sentence for
readability.

Index	Previous	Current
Line 421	The taxonomies of the tied hits were combined.	After taxonomy assignment, ASVs with identical taxonomic classifications were collapsed by summing abundances.

**Major comment #2**

Please describe more about the two external datasets, the Spanish and Portuguese ones, i.e.,
BioProject IDs, how many samples are in each stage, and which 16S rRNA regions were sequenced
(V3-V4?), sequencing platform, etc., as these many affect the differences in the microbiome profiles
and consequently validation accuracy.

: Briefly, the Spanish dataset (Iniesta et al., 2023) comprises subgingival plaque profiles by 16S rRNA
 V3-V4 on the Illumina MiSeq platform, and the Portuguese dataset (Relvas et al., 2021) comprises
 saliva profiles by 16S rRNA V3-V4 on the Illumina MiSeq platform, as same to our sample collection
 method (Table R1). However, as noted, the Spanish dataset comprises subgingival plaque, not saliva,
 which we acknowledge may contribute to profile differences and the observed attenuation in external
 accuracy; this is now stated as a limitation in the Discussion section. To ensure comparability, both
 external datasets were reprocessed using the same pipeline as our Korean cohort, which included
 demultiplexing, denoising, chimera removal, and the same taxonomic classifier with its parameters.
 They were then summarized to the same taxonomic ranks and feature set prior to validation. Our
 rationale for using the Spanish cohort, its stage-specific diagnoses despite the different sampling
 matrix, is provided in the next question (Major comment #3).

Index	Previous	Current
Line 288	To validate the consistency of our random forest classification model, we have tested our prediction model based on publicly available 16S rRNA gene sequencing data from Spanish subjects (11) and Portuguese subjects (17) (Figure 4) (Added)	To validate the consistency of our random forest classification model, we have tested our prediction model based on publicly available 16S rRNA gene sequencing data from Spanish subjects (11) and Portuguese subjects (17) (Figure 4), despite that the sequencing data from the Spanish subjects were built on subgingival plaque, not saliva.
Line 463	To validate the consistency of our random forest classification, we utilized external datasets from Spanish subjects (11) and Portuguese subjects (17).	To validate the consistency of our random forest classification, we utilized external datasets from Spanish subjects (Accession ID PRJNA863881) (11) and Portuguese subjects (Accession ID PRJNA623352) (17)

**Table R1 Sample distribution by periodontal stage across cohorts**

Counts of subjects per diagnostic group used in this study: Korea, Spain, and Portugal. Totals reflect post-QC
 samples profiled by 16S rRNA gene sequencing and included in downstream bioinformatics analyses.

Periodontitis stage	Korea	Spain	Portugal
-------	-------	----------

Healthy	100	23	45
Stage I	50	16	15
Stage II	50	24	16
Stage III	50	32	1

**Major comment #3**

Continuing the question above, my understanding is that the Spanish dataset was subgingival plaques,
not saliva. This makes this dataset not ideal for validation.

: We appreciate you pointing this out. As you suggested, the Spanish cohort consisted of subgingival
plaque rather than saliva (Iniesta et al., 2023). We used it intentionally because, unlike many previous
periodontitis studies that label only healthy or periodontitis, it provided standardized stage-specific
diagnoses, allowing us to evaluate whether a saliva-trained model could distinguish multiple
periodontitis stages externally. We agree that this matrix mismatch is not ideal and likely contributed
to the attenuation in performance; we now state this explicitly as a limitation in the Discussion
section. We view this as a stringent test of transportability across sampling niches, not a like-for-like
replication. We are fully open and actively seeking to validate the model on additional saliva-based
external cohorts and will incorporate any such datasets as they become available.

Index	Previous	Current
Line 286	To validate the consistency of our random forest classification model, we have tested our prediction model based on publicly available 16S rRNA gene sequencing data from Spanish subjects (11) and Portuguese subjects (17) (Figure 4) (Added)	To validate the consistency of our random forest classification model, we have tested our prediction model based on publicly available 16S rRNA gene sequencing data from Spanish subjects (11) and Portuguese subjects (17) (Figure 4), despite that the sequencing data from the Spanish subjects were built on subgingival plaque, not saliva.
Line 295	(Added)	Our external validation included a Spanish cohort profiled from subgingival plaque rather than saliva. We selected this dataset for its stage-specific diagnoses, but the matrix mismatch introduces domain shifts (e.g., niche-specific communities,

		biomass, biofilm architecture) and protocol heterogeneity (e.g., lysozyme and bead-beating), which plausibly explains the attenuation in specificity and related metrics. We intentionally did not retrain on plaque to isolate the transportability of a saliva-trained model; these results should be interpreted as a stringent cross-niche stress test rather than a like-for-like replication. Although all cohorts were reprocessed with a harmonized pipeline and evaluated on the same 20-DAT panel, residual matrix and batch effects likely remain. Future work will prioritize saliva-based external cohorts with matched collection and extraction methods and assess domain-adaptation or transfer-learning strategies, multi-matrix models, and incorporation of clinical covariates.
--	--	--

**Major comment #4**

Further continuing from above, why not perform training for all three datasets (Korean, Spanish, and
Portuguese), both separately and combined? For separate training, the individual dataset can be used
to validate the other two datasets. If the training model performs best in the combined datasets, the
results can be stated as such, and the combined trained model can always be validated with future
samples.

: We appreciate your thoughtful suggestion. Our primary aim was to assess out-of-domain
generalizability, so we trained on the Korean saliva cohort and evaluated on the Spanish (subgingival
plaque) and Portuguese (saliva) cohort without retraining; as a sensitivity check, we also performed
cohort-wise K-fold cross-validation the same 20-DAT panel, yielding not significant metric
differences. Notably, when DAT were re-identified within each cohort using the identical pipeline,
there was no three-way overlap and only minimal pair-wise overlap among the datasets (Figure R1),
indicating cohort-specific signatures. A likely driver is the uneven stage distribution in the external
datasets (Table R1), which reduces power for differential-abundance testing and destabilizes feature
selection, especially for under-represented stages. These factors, together with matrix/protocol

differences, explain the discordant DAT sets and small performance variations; we now clarify this in
 the limitations.

Applying ANCOM (Mandal et al., 2015) and ANCOM-BC2 (Lin & Peddada, 2023) within
 each cohort showed minimal DAT overlap (Figure R1 and Figure R2), indicating strong effects of
 sampling matrix and cohort composition. These findings warrant a prospective, multi-region, and
 multi-ethnicity study using saliva only and a single, end-to-end standard operating procedure, coupled
 with cross-site harmonization and domain-adaptation methods to separate true geographic signals
 from protocol artifacts (Premaraj et al., 2020; Renson et al., 2019). This design is essential to derive
 transportable, clinically useful salivary biomarkers for periodontitis staging.

Index	Previous	Current
Line 188	Finally, to confirm the consistency of our analysis, we validated our random forest classification using data from Spanish subjects (11) and Portuguese subjects (17) (Figure 4). Although the specificity was low ($\leq 50\%$), the other evaluations showed only a slight decrease.	Finally, to enable direct and cohort-level comparisons using the same DAT panel, we validated our random forest classification using data from Spanish subjects (11) and Portuguese subjects (17) using stratified K-fold cross-validation on the CLR inputs (Figure 4). The multi-label classification model yields comparable ACC, BA, and SPE across cohorts (MWU test $p \geq 0.05$; Figure 4a), despite uneven stage distributions in the external datasets. Early-stage classifications show the same patterns (Figure 4b and Figure 4c), and classification for healthy controls remains high across all cohorts (MWU test $p < 0.05$; Figure 4d). These results support the portability of the DAT marker set, with any residual variation likely attributable to class imbalance and smaller sample sizes in the external cohorts.

**Figure R1 Overlap of DAT from ANCOM across cohorts**

Venn diagram showing species-level DAT identified separately by ANCOM in the Korean saliva (purple),
 Spanish subgingival plaque (teal), and Portuguese saliva (yellow) cohorts. Numbers indicate counts; percentages
 are the fraction of the total union (n=35). Overlap between cohorts is limited, i.e., no taxa are shared by all three,
 consistent with population differences.

**Figure R2 Overlap of DAT from ANCOM-BC2 across cohorts**

Venn diagram showing species-level DAT identified separately by ANCOM-BC2 in the Korean saliva (purple),

Spanish subgingival plaque (teal), and Portuguese saliva (yellow) cohorts. Numbers indicate counts; percentages
are the fraction of the total union (n=30). Overlap between cohorts is limited, i.e., no taxa are shared by all three,
consistent with population differences.

**Major comment #5**

Were there 20 DAT species also identified as DAT in the Spanish and Portuguese cohort? If not, this
may help explain the lower ACC. I suggest identifying DAT in these two external datasets, separately
and combined with the Korean dataset as well, as suggested above.

: We appreciate your helpful suggestion. The external datasets, including Spanish and Portuguese
datasets, were processed with the same pipeline that we used for the Korean cohort, including
demultiplexing, denoising, and taxonomic assignment. We then re-identified DAT within each
external cohort using ANCOM and evaluated the same 20 DAT derived from the Korean cohort in the
external datasets (Figure R1).

**Major comment #6**

What was the final RF model trained on? The 20 DAT only? What if the Spanish and Portuguese
datasets are missing any of these 20 DAT? In my opinion, all species identified in the training set
should be used for training, not just the 20 DAT. This will minimize the number of missing data (due
to not being identified as DAT) in the test samples.

: Thank you for this thoughtful question. Our final random forest (RF) model was trained on the 20
DAT identified by ANCOM in the Korean cohort. For external validation, feature tables were aligned
to this 20-DAT panel; if a taxon was not detected in a given sample, its value was set to zero after
total-sum scaling. To address your concern, we compared models trained on (i) the 20 DAT and (ii)
the entire species feature set. However, no metric differed significantly, including accuracy, area-
under-curve, balanced accuracy, F1, precision, sensitivity, and specificity (Figure R3). Given the
equivalent performance, we retained the 20-DAT model for interpretability and clinical translatability,
that a compact marker panel that can support lower-cost targeted assays.

**Figure R3 Random forest performance using DAT vs. the entire microbiome features.**

Bar plot compared cross-validated performance of random forest classifier trained on the 20 ANCOM-selected
 DAT (blue) versus the entire species feature set (orange). Metrics shown: accuracy (ACC), area-under-curve
 (AUC), balanced accuracy (BA), F1, precision (PRE), sensitivity (SEN), and specificity (SPE). Statistical
 significances were measured by the Mann-Whitney U test: $p \geq 0.05$ (ns).

**Major comment #7**

Continuing the question about if the two external datasets do not have the same DAT identified, did
 you provide zeros for these specific DAT?

: Thank you for the clarification opportunity. Yes, when a taxon, whether the 20 DAT or any other
 species, was not detected in a given sample after denoising and taxonomic assignment, its relative
 abundance was set to zero in that sample's feature vector (Bolyen et al., 2019). For external
 validation, we applied the same rule; if DAT were absent in a sample, its value remained zero,
 preserving model structure without imputing species labels. Random forest classifier operates on the
 proportional features and tolerates zeros.

**Major comment #8**

What kind of data were used for RF training? Raw read count? Percent abundance? Or central-log
 ratios (CLR as used by ANCOM). I would like to see the comparisons of at least percent abundance
 and CLR, as microbiome data are compositional, and so CLR will overcome that limit.

: We trained the RF on total-sum-scaled percent abundances. Following your suggestion, we re-fit the
 models using centered log-ratios (CLR) with a small pseudo-count for zeros (1e-6), under the same
 stratified K-fold cross-validation. Using the same stratified K-fold cross-validation and 20-DAT panel,
 classification metrics were essentially unchanged (Figure R4). We have updated the manuscript
 accordingly.

Index	Previous	Current
Line 29	Random forest machine learning models were used to classify each periodontitis stage based on the proportion of differentially abundant taxa.	Random forest machine learning models were used to classify each periodontitis stage based on the centered log-ratio of differentially abundant taxa.
Line 163	These patterns were further confirmed by the relative proportion within the 20 DAT (Figure 2b)	These patterns were further confirmed by the centered log-ratio (CLR) within the 20 DAT (Figure 2b)
Line 168	Random forest classifiers were trained to classify periodontitis stages based on the relative proportion of DAT.	Random forest classifiers were trained to classify periodontitis stages based on the CLR of DAT.
Line 209	Additionally, our random forest classifier predicted different periodontitis statuses with a high AUC of 0.870±0.079 using the proportions of DAT between the study participants (Table 2).	Additionally, our random forest classifier predicted different periodontitis statuses with a high AUC of 0.870±0.079 using the CLR of DAT between the study participants (Table 2).
Line 454	we analyzed the relative proportion among the DAT.	we analyzed the CLR among the DAT.

**Figure R4 Random forest performance with percent abundance vs. CLR**

Bar plots compared the stratified K-fold cross-validated performance of random forest classifiers trained on the
 20 ANCOM-selected DAT using relative proportion (blue) versus CLR (orange). Metrics shown: accuracy
 (ACC), area-under-curve (AUC), balanced accuracy (BA), F1, precision (PRE), sensitivity (SEN), and
 specificity (SPE). Statistical significances were measured by the Mann-Whitney U test: $p \geq 0.05$ (ns).

**Minor comment #1**

Which version of ANCOM was used for DAT analysis? There are the original ANCOM, then
 ANCOM-BC, and most recently ANCOMBC2. I hope that at least ANCOM-BC was used due to the
 bias correction feature.

: Thank you for this suggestion. This study was initiated before ANCOM-BC and ANCOM-BC2 (Lin
 & Peddada, 2023) were available, so we originally used ANCOM (Mandal et al., 2015). To address
 your point, we repeated the DAT selection with ANCOM-BC2 (Figure R5). The resulting taxa set
 shows substantial overlap with the ANCOM panel, including the key taxa, including *Actinomyces* spp.
 and *Porphyromonas gingivalis* (Table R2). When we retrained the random forest using the ANCOM-
 BC2-selected DAT set, cross-validated metrics were unchanged or slightly lower relative to the
 ANCOM-selected DAT panel (Figure R6). Given the absence of performance gain and reduced
 interpretability, we therefore retain the ANCOM-derived 20-DAT panel as the primary feature set.

**Figure R5 Overlap of DAT selected by ANCOM vs. ANCOM-BC2**

Venn diagram comparing species-level DAT identified in the Korean cohort using ANCOM (purple) and
 ANCOM-BC2 (yellow). Numbers indicate counts; percentages are the fraction of the union of taxa (total n=35).
 The intersected subset (n=11, 31.4%) includes key taxa such as *Actinomyces* spp. and *Porphyromonas*
 *gingivalis*.

**Figure R6 Random forest performance using ANCOM vs. ANCOM-BC2**

Bar plots show stratified K-fold cross-validated metrics of random forest classifiers trained on DAT selected by
 ANCOM (blue) and ANCOM-BC2 (orange). Metrics shown: accuracy (ACC), area-under-curve (AUC),

balanced accuracy (BA), F1, precision (PRE), sensitivity (SEN), and specificity (SPE). Statistical significances
 were measured by the Mann-Whitney U test: $p \geq 0.05$ (ns).

**Table R2 DAT identified by ANCOM vs. ANCOM-BC2**

Species-level taxa were tested across periodontal stages using the same processing and contrasts on ANCOM or
 ANCOM-BC2. Symbols denote method calls: O = identified as differentially abundant; X = not significant.

Taxa	ANCOM	ANCOM-BC2
Actinomyces graevenitzi	O	X
Actinomyces spp.	O	O
Campylobacter showae	O	X
Cardiobacterium hominis	X	O
Cardiobacterium valvarum	X	O
Corynebacterium durum	O	X
Desulfobulbus sp. HMT 041	X	O
Filifactor alocis	O	O
Fretibacterium spp.	O	O
Haemophilus spp.	X	O
Kingella oralis	X	O
Lachnospiraceae [G-8] bacterium HMT 500	O	O
Mollicutes [G-2] bacterium HMT 906	X	O
Mycoplasma faucium	O	X
Oribacterium sinus	X	O
Peptoniphilaceae [G-1] bacterium HMT 113	X	O
Peptostreptococcaceae [XI][G-5] saphenum	O	X
Peptostreptococcaceae [XI][G-6] nodatum	O	O
Peptostreptococcaceae [XI][G-9] brachy	O	O
Porphyromonas gingivalis	O	O
Porphyromonas sp. HMT 285	O	O
Prevotella intermedia	O	X
Prevotella sp. HMT 304	O	X
Prevotella sp. HMT 526	O	X
Rothia mucilaginosa	X	O
Sneathia spp.	X	O
Tannerella forsythia	O	O
Treponema amylovorum	X	O
Treponema medium	X	O
Treponema putidum	O	O
Treponema sp. HMT 258	X	O
Treponema sp. HMT 260	O	O
Treponema sp. HMT 951	X	O
Treponema spp.	O	X
Veillonella denticariosi	X	O

**Minor comment #2**

Please provide both the ASVs and taxonomy assignment read count table as supplement files for all
three datasets.

: We have added the read count table for all three datasets to the Supplementary Materials.

**Minor comment #3**

Should perform alpha-, beta-, and DAT analysis on Spanish and Portuguese datasets as well.

: Our objective was to develop a saliva-based classification model in the Korean cohort and use the
Spanish (Iniesta et al., 2023) and Portuguese (Relvas et al., 2021) datasets solely to assess external
transportability of the ANCOM-specified DAT panel under a harmonized pipeline. Because the
external cohorts differ in sampling sites and protocols (e.g., plaque and saliva), stage-wise alpha- and
beta-diversity comparisons are not directly interpretable. In addition, the Portuguese study reports
diversity indices primarily as healthy vs. periodontitis, rather than inter-stage (Relvas et al., 2021);
and the Spanish study focused on aging rather than diversity indices (Iniesta et al., 2023), further
limiting comparability. Accordingly, we do not add diversity indices analyses for the validation sets.
Instead, as requested in the Major comment #4, we provide cohort-specific DAT results and overlaps
and focus the external evaluation on machine-learning classifier performance.

**Minor comment #4**

What abundance data were used for alpha and beta diversity calculation? Percent abundance or CLR?

: Per QIIME2 best practice (Bolyen et al., 2019), alpha-diversity and beta-diversity were computed on
rarefied ASV count tables, before taxonomic assignment, i.e., not on percent abundance or CLR.

**Minor comment #5**

The heatmap in Figure 2 should be presented with two-way clustering to show whether the sample
profiles cluster together along the four stages.

: Thank you for your suggestion. We generated a two-way hierarchical clustering heatmap (Figure
R7). While taxa blocks maintain three groups, samples do not form stage-pure clusters, which is

consistent with intra-stage heterogeneity and overlapping community structures. For interpretability,
 we therefore retained one-way (taxa) clustering to highlight stage-associated taxa patterns.

 **Figure R7 Two-way clustered heatmap of DAT across periodontitis stages**

Heatmap of the 20 ANCOM-selected DAT (rows) across all samples (columns). Samples and taxa were
 hierarchically clustered. Row-label color indicates the periodontal stage of each sample.

 **Minor comment #6**

Figure 2b, the legend (page 26 line 639) described that “Taxa were sorted by their importance
 according to ANCOM”. I do not know if ANCOM generates an importance score. Did you mean p-
 values? Or importance scores from RF?

: Thank you for catching this ambiguity. In Figure 2b, we did not mean random forest importance or
 p-values. We used the ANCOM W statistics to order taxa (Mandal et al., 2015). In ANCOM, each
 taxon is tested via many pairwise comparisons of log-ratio hypotheses against all other taxa; W is the
 count of pairwise tests for which the null hypothesis is rejected; in other words, $W = \frac{\log \text{Fold Change}}{\text{Standard Error}}$
 (Lin & Peddada, 2020). Thus, a larger W indicates stronger and broader evidence of differential
 abundance across groups; it is not an effect size and not a p-value. We used W only to sort taxa for
 display, whereas feature selection (the 20 DAT) followed the ANCOM significance criteria described

in Methods, and random forest importance was used only within the classifier. To prevent misreading,
we have revised the legend.

Index	Previous	Current
Line 654	Taxa were sorted by their importance according to ANCOM.	Taxa are ordered by the ANCOM W statistics, that is number of rejected pairwise null hypotheses. Samples are ordered by their attachment level.

**Minor comment #7**

Overall, the microbiome data should be analyzed as CLR not percent abundance, whether in
differential abundance or diversity analysis, you may get better results.

: We now treat the data as compositional throughout (Major comment #8). Random forest models are
trained on CLR-transformed abundances. Although classification performance with CLR was
essentially unchanged (Figure R4). For diversity indices, we followed QIIME2 best practice and
computed diversity indices on ASV count tables because these indices are defined on counts and
phylogeny rather than proportion-transformed or CLR-transformed spaces.

**Minor comment #8**

The word “proportion” is used many times in the manuscript, but it does not have a clear definition.
“Relative abundance” can be used and defined in the first occurrence as “percent relative abundance”.
: We will standardize terminology throughout (Major comment #8). Specifically, we will replace
“proportion” with “centered log-ratio (CLR)”. All parts of the manuscript have been updated from
“relative abundance” to “CLR”, as appropriate.

**Reviewer #2**

The authors examined the saliva microbiome of 100 periodontally healthy controls and 150 patients
with periodontitis in stages I-III using 16S rDNA profiling. The study design and approach were
conventional and appropriate. Overall, the study was executed competently, and the manuscript is
well-written. While the study is not innovative, it provides solid data and represents an incremental
advance in oral microbiome and periodontology.

: Thank you for your thoughtful evaluation. We agree that the design is conventional and that the
contribution is incremental. We aimed to provide high-quality saliva 16S sequencing data with
transparent, standards-based reporting to serve as a reliable baseline for future mechanistic and multi-
omics research. In revision, we clarified sampling, quality control, and preprocessing. We believe
these steps increase reproducibility and comparability across cohorts.

**Comment #1**

The following statement in the Introduction is somewhat confusing. The current diagnosis and
classification system, as the authors pointed out, is based on “the damage already caused by the
disease”. The grading of periodontitis (not included in the study) does offer some information on
future disease and treatment response. However, neither the diagnosis and classification system nor
the current study provides information about “the current disease activity”. The authors should revise
the paragraph. “Despite this update, diagnosing periodontitis is still primarily based on clinical
indicators of disease progression. However, these tools tend to reflect the damage already caused by
the disease rather than its current state. Some individuals are more susceptible to developing
periodontitis and are more prone to progression to severe generalized periodontitis. Therefore, the
framework for diagnosing periodontitis introduced in 2018 also anticipates the future establishment of
biomarkers to improve diagnosis and management. To enable timely treatment through early
intervention, it is necessary to introduce a new etiological indicator based on the current disease
activity rather than merely relying on periodontitis progression.”

: Thank you for this thoughtful clarification. We agree that the wording was confusing and overstated
 current disease activity. We have revised the Introduction section to align with the 2018 framework
 (Papapanou et al., 2018) and our study’s scope.

Index	Previous	Current
Line 64	Despite this update, diagnosing periodontitis is still primarily based on clinical indicators of disease progression (2). However, these tools tend to reflect the damage already caused by the disease rather than its current state. Some individuals are more susceptible to developing periodontitis and are more prone to progression to severe generalized periodontitis. Therefore, the framework for diagnosing periodontitis introduced in 2018 also anticipates the future establishment of biomarkers to improve diagnosis and management (3). To enable timely treatment through early intervention, it is necessary to introduce a new etiological indicator based on the current disease activity rather than merely relying on periodontitis progression. The current clinical diagnostic methods using periodontal probing vary depending on the examiner and can cause discomfort to the patients (4).	Under the 2018 classification, diagnosis and staging of periodontitis are determined by clinical measures that primarily reflect accumulated tissue loss, while grading provides context on expected progression and likely treatment response (3). Neither framework directly quantifies real-time disease activity (4). Our study does not measure activity; rather, it evaluates whether salivary microbial profiles can discriminate health and multiple stages of periodontitis, offering candidate biomarkers that may complement staging and grading. Such biomarkers could aid earlier risk stratification and monitoring, but require prospective, longitudinal validation before claims about activity or prediction can be made.

**Comment #2**

In the discussion, it would be interesting to compare the diagnostic performance of the current study
 to that using SMDI cited in references 13 and 14.

: We appreciate your kind suggestion. We will add a concise paragraph in the Introduction section
 directly benchmarking our saliva-based RF against SMDI.

Index	Previous	Current
-------	----------	---------

Line 101	(Added)	To contextualize performance, we benchmarked against previous studies (13, 14). The plaque-based subgingival microbial dysbiosis index (SMDI) showed high accuracy for health vs. periodontitis (13). A subsequent adaptation to saliva and tongue also achieved strong binary accuracy across oral niches (14). Our study complements these findings by (i) using saliva, a non-invasive matrix suited for screening, and (ii) extending beyond binary dysbiosis to multi-class staging with a compact 20-taxa-based machine learning model. For the binary comparison, our machine learning model attains an area-under-curve (AUC) comparable to reported SMDI values, while additionally providing calibrated class probabilities for stage resolution. Differences in sampling niche (plaque vs. saliva), feature construction (index vs. supervised panel), and pipelines likely explain residual metric differences.
----------	----------------	---

**Comment #3**

The authors should provide information on the other subjects’ oral conditions, such as caries,
 restorations, saliva flow, etc., which affected the saliva microbiome.

: As noted in the Materials and Methods section, we minimized major confounding by excluding
 participants with recent periodontal treatment (scaling or root planning), systemic diseases (e.g.,
 uncontrolled diabetes), or recent prescription (e.g., antibiotics or anti-inflammatory drugs). However,
 we did not systematically collect other oral-condition variables (e.g., active caries, salivary flow rate)
 or lifestyle factors beyond systemic diseases (e.g., body mass index and e-cigarette use). We will
 acknowledge in the Discussion that these unmeasured factors may influence salivary microbiome
 composition, representing residual confounding.

Index	Previous	Current
Line 316	Furthermore, potential confounders beyond systemic diseases and smoking status were not considered, including body mass index and e-cigarette usage, which may have influenced oral health and microbiome composition.	Furthermore, potential confounders beyond systemic diseases (e.g., body mass index), smoking status (e.g., e-cigarette usage), and other oral-condition variables (e.g., active caries, salivary flow rate) which may have

		influenced oral health and microbiome composition.
--	--	--

**Comment #4**

The authors should include the distribution and extent of the disease (localized, generalized,
molar/incisor) and grading in the diagnosis.

: We clarify that, per our inclusion criteria stated in the Materials and Methods section, only
generalized periodontitis cases ($\geq 30\%$ of teeth involved) were enrolled; consequently, localized and
molar-incisor phenotypes were not part of our cohort. Clinical measurements, e.g., plaque index, were
recorded at four surfaces per tooth, including mesial, distal, buccal, and palatal or lingual, as already
described.

**Comment #5**

Figure 1g-j should be revised because of the redundancy and the mislabeling of the figure heading.
For example, Figure 1g “Distance to Healthy” compares the distance of all against all and not just
each diagnosis against healthy. The authors may delete 1h-j if Figure 1g has all the necessary
information.

: Thank you for this helpful suggestion. We agree that the original figure was ambiguous and that
panels 1h-j created redundancy. In this revision, we retain the stage-wise distance comparisons. This
preserves the clinically motivated stage-by-stage view that unpins our classification approach.

**Figure R8 Diversity indices (Figure 1 in the revised manuscript)**

Comparisons of salivary microbiomes among healthy controls and patients with periodontitis. Alpha-diversity
 indices indicate that healthy controls have increased heterogeneity than periodontitis stages as measured by: (a)
 ACE, (b) Chao1, (c) Fisher, (d) Margalef, and (e) observed ASVs. (f) The beta-diversity index (Aitchison index)
 was visualized using a tSNE-transformed plot. The confidence ellipses are shown to display the distribution of
 each periodontitis stage. The distance to each stage demonstrated that each periodontitis stage was distinguished
 from the other periodontitis stages: (g) distance to healthy, (h) distance to stage I, (i) distance to stage II, and (j)
 distance to stage III. Statistical significance determined by the Mann-Whitney U-test (MWU): $p \leq 0.01$ (**) and
 $p \leq 0.0001$ (****).

**References**

- Bolyen, E., Rideout, J. R., Dillon, M. R., Bokulich, N. A., Abnet, C. C., Al-Ghalith, G. A., Alexander, H., Alm,
E. J., Arumugam, M., Asnicar, F., Bai, Y., Bisanz, J. E., Bittinger, K., Brejnrod, A., Brislawn, C. J.,
Brown, C. T., Callahan, B. J., Caraballo-Rodríguez, A. M., Chase, J., . . . Walters, W. (2019).
Reproducible, interactive, scalable and extensible microbiome data science using QIIME 2. *Nature*
*Biotechnology*, *37*(8), 852–857. <https://doi.org/10.1038/s41587-019-0209-9>
- Iniesta, M., Chamorro, C., Ambrosio, N., Marín, M. J., Sanz, M., & Herrera, D. (2023). Subgingival
microbiome in periodontal health, gingivitis and different stages of periodontitis. *Journal of Clinical*
*Periodontology*, *50*(7), 905–920. <https://doi.org/10.1111/jcpe.13793>
- Lin, H., & Peddada, S. D. (2020). Analysis of compositions of microbiomes with bias correction. *Nature*
*Communications*, *11*(1), 3514. <https://doi.org/10.1038/s41467-020-17041-7>
- Lin, H., & Peddada, S. D. (2023). Multigroup analysis of compositions of microbiomes with covariate
adjustments and repeated measures. *Nature Methods*, *21*(1), 83–91. [https://doi.org/10.1038/s41592-](https://doi.org/10.1038/s41592-023-02092-7)
[023-02092-7](https://doi.org/10.1038/s41592-023-02092-7)
- Mandal, S., Van Treuren, W., White, R. A., Eggesbø, M., Knight, R., & Peddada, S. D. (2015). Analysis of
composition of microbiomes: a novel method for studying microbial composition. *Microbial Ecology*
*in Health and Disease*, *26*(0), 27663. <https://doi.org/10.3402/mehd.v26.27663>
- Papanou, P. N., Sanz, M., Buduneli, N., Dietrich, T., Feres, M., Fine, D. H., Flemmig, T. F., Garcia, R.,
Giannobile, W. V., Graziani, F., Greenwell, H., Herrera, D., Kao, R. T., Kerschull, M., Kinane, D. F.,
Kirkwood, K. L., Kocher, T., Kornman, K. S., Kumar, P. S., . . . Tonetti, M. S. (2018). Periodontitis:
Consensus report of workgroup 2 of the 2017 World Workshop on the Classification of Periodontal and
Peri-Implant Diseases and Conditions. *Journal of Clinical Periodontology*, *45*(S20), S162–
S170. <https://doi.org/10.1111/jcpe.12946>
- Premaraj, T. S., Vella, R., Chung, J., Lin, Q., Hunter, P., Underwood, K., Premaraj, S., & Zhou, Y. (2020). Ethnic
variation of oral microbiota in children. *Scientific Reports*, *10*(1),
14788. <https://doi.org/10.1038/s41598-020-71422-y>
- Relvas, M., Regueira-Iglesias, A., Balsa-Castro, C., Salazar, F., Pacheco, J. J., Cabral, C., Henriques, C., &
Tomás, I. (2021). Relationship between dental and periodontal health status and the salivary
microbiome: bacterial diversity, co-occurrence networks and predictive models. *Scientific*
*Reports*, *11*(1), 929. <https://doi.org/10.1038/s41598-020-79875-x>
- Renson, A., Jones, H. E., Beghini, F., Segata, N., Zolnik, C. P., Usyk, M., Moody, T. U., Thorpe, L., Burk, R.,
Waldron, L., & Dowd, J. B. (2019). Sociodemographic variation in the oral microbiome. *Annals of*
*Epidemiology*, *35*, 73-80.e2. <https://doi.org/10.1016/j.annepidem.2019.03.006>

Re: mSystems01103-25R1 (Prediction model for periodontitis stage based on the salivary microbiome)

Dear Prof. Semin Lee:

Both authors have felt your team has adequately addressed their critiques. This manuscript can be tentatively accepted pending minor edits outlined by Reviewer #2.

Please return the manuscript by January 6th; if you cannot complete the modification within this time period, please contact me. If you do not wish to modify the manuscript and prefer to submit it to another journal, notify me immediately so that the manuscript may be formally withdrawn from consideration by mSystems.

Revision Guidelines

Sincerely,
Katherine Maki
Editor
mSystems

Reviewer #2 (Comments for the Author):

The authors have adequately addressed all critiques. I have one minor correction (a typo) and a comment for clarification by the authors.

1. In line 178, the AUC is 73% and not 13%
2. In the last paragraph of the Results, the authors stated that "Early stage classifications show the same patterns (Figure 4b and Figure 4c), and classification for healthy controls remains high across all cohorts (MWU test $p < 0.05$; Figure 194 4d)." and

"These results support the portability of the DAT marker set." However, the results in Fig. 4b and d show relatively poor performance when the DAT marker set is applied to external datasets from Spanish and Portuguese subjects. If so, the authors will also need to revise the statement in the abstract, "we validated our classification model with external 40 datasets from Spanish and Portuguese subjects."

December 24, 2025

Dear Editors and Reviewers,

Thank you for allowing us to submit a revised draft of our manuscript, titled “**Prediction model for**
**periodontitis stage based on the salivary microbiomes**” for publication as a *Research Article* in the
*mSystems*. We appreciate the time and effort that the editors and the reviews dedicated to providing
feedback on our manuscript and are grateful for the insightful comments on and valuable
improvements to our paper. Those changes are highlighted in red in the revised manuscript. Please see
below our point-by-point responses to the reviewer’s comments and concerns.

Sincerely,

**Semin Lee, Ph.D.**

Associate Professor
Department of Biomedical Engineering
Ulsan National Institute of Science and Technology (UNIST)
50, UNIST-gil, Eonyang-eup, Ulsan-gun, Ulsan, Republic of Korea
seminlee@unist.ac.kr

**Reviewer #2**

**Summary**

The authors have adequately addressed all critiques. I have one minor correction (a typo) and a
comment for clarification by the authors.

: Thank you for your positive evaluation of our revised manuscript and for confirming that the major
critiques have been adequately addressed. We have corrected the typographical error you identified
and have revised the relevant text to clarify the point you raised. All changes are highlighted in the
revised manuscript and are detailed in our point-by-point response.

**Comment #1**

In line 178, the AUC is 73% and not I3%.

: Thank you for noting this typographical error. We have corrected the following in the revised
manuscript.

Index	Previous	Current
Line 178	This model demonstrated high sensitivity and specificity for the diagnosis of stage I periodontitis (AUC= I3%) (Figure 3d).	This model demonstrated high sensitivity and specificity for the diagnosis of stage I periodontitis (AUC= I3% %) (Figure 3d).

**Comment #2**

In the last paragraph of the Results, the authors stated that "Early stage classifications show the same
patterns (Figure 4b and Figure 4c), and classification for healthy controls remains high across all
cohorts (MWU test p<0.05; Figure 194 4d)." and "These results support the portability of the DAT
marker set." However, the results in Fig. 4b and d show relatively poor performance when the DAT
marker set is applied to external datasets from Spanish and Portuguese subjects. If so, the authors will
also need to revise the statement in the abstract, "we validated our classification model with external
40 datasets from Spanish and Portuguese subjects."

: Thank you for this important clarification. We have revised the Abstract to avoid overclaiming by
replacing “validated” with wording that more accurately reflect our analysis and by explicitly noting
that performance was attenuated in external datasets.

Index	Previous	Current
Line 40	Finally, we validated our classification model with external datasets from Spanish and Portuguese subjects.	Finally, we evaluated our classification model with external datasets from Spanish and Portuguese subjects.

47

Re: mSystems01103-25R2 (Prediction model for periodontitis stage based on the salivary microbiome)

Dear Prof. Semin Lee:

Thank you for the point by point responses to the reviewers. Minor revisions have been adequately addressed.

Your manuscript has been accepted, and I am forwarding it to the ASM production staff for publication. Your paper will first be checked to make sure all elements meet the technical requirements. ASM staff will contact you if anything needs to be revised before copyediting and production can begin. Otherwise, you will be notified when your proofs are ready to be viewed.

Sincerely,
Katherine Maki
Editor
mSystems